# Genetic evidence for amlodipine's protective role in gastroesophageal reflux disease: A focus on *CACNB2*

**Liuzhao Zhang[1], Quanwang Chu[1], Shuyue Jiang[1], Bo Shao[2]\***

1 Department of Critical Care Medicine, Anhui Jing'an Medicine Hospital, Hefei, China, 2 Department of Pathology, Anhui Provincial Children's Hospital, Hefei, China

\* shaobo5988@163.com

## Abstract

### Objective

This study aims to elucidate the causal relationship between genetically predicted amlodipine use and the risk of gastroesophageal reflux disease (GERD) using a bidirectional Mendelian Randomization (MR) approach and to explore the underlying genetic and molecular mechanisms through functional enrichment analysis and the construction of a competing endogenous RNA (ceRNA) network.

### Methods

Publicly available GWAS datasets from the Neale Lab consortium were used, including data on amlodipine (13,693 cases, 323,466 controls) and GERD (14,316 cases, 322,843 controls). Genome-wide significant SNPs were selected as instrumental variables and clustered by linkage disequilibrium. MR analysis was conducted using R software with all five methods. Sensitivity analyses assessed pleiotropy and heterogeneity. Drug target genes were analyzed using GO and KEGG pathways. GeneMANIA was used for network visualization, and a ceRNA network was constructed with Cytoscape. Differential gene expression analysis on GERD-related datasets from GEO validated the findings.

### Results

The MR analysis indicated a significant negative association between genetically predicted amlodipine use and GERD risk (IVW OR = 0.872, 95% CI = 0.812–0.937, P = 0.0002). Sensitivity analyses confirmed the robustness of these findings, showing no evidence of pleiotropy or heterogeneity. The enrichment analysis identified key biological processes and pathways involving calcium ion transport and signaling. The ceRNA network highlighted core targets such as CACNB2, which were further validated by differential expression analysis intersecting drug target genes with GERD-related gene expression changes.

**Data Availability Statement:** All relevant data are within the paper and its Supporting Information files.

**Funding:** The author(s) received no specific funding for this work.

**Competing interests:** The authors have declared that no competing interests exist.

## Conclusion

This study provides robust evidence of a protective effect of amlodipine against GERD, supported by genetic and molecular analyses. The findings suggest that calcium channel blockers like amlodipine could be repurposed for GERD treatment. The identification of CACNB2 and other core targets in the ceRNA network offers novel insights into the pathophysiology of GERD and potential therapeutic targets, paving the way for personalized medicine approaches to improve patient outcomes.

## Introduction

Gastroesophageal reflux disease (GERD) is a chronic condition characterized by the backflow of stomach contents into the esophagus, causing symptoms such as heartburn, regurgitation, and, in severe cases, esophageal mucosal damage [1, 2]. GERD is prevalent worldwide, affecting approximately 20% of the Western population and a significant portion of individuals in other regions [3]. Despite the availability of effective pharmacological treatments like proton pump inhibitors (PPIs) and lifestyle modifications aimed at reducing acid reflux, a considerable number of patients continue to experience persistent or recurrent symptoms [4–6]. This highlights the complexity of GERD's pathophysiology and the need for more effective or adjunctive therapeutic strategies. Amlodipine, a long-acting dihydropyridine calcium channel blocker, is widely prescribed for managing hypertension and angina pectoris [7]. It functions primarily by inhibiting calcium ion influx into vascular smooth muscle and cardiac muscle cells, leading to vasodilation and reduced blood pressure [8]. However, the influence of amlodipine on gastrointestinal motility and the lower esophageal sphincter (LES) function has been a topic of ongoing research and debate [9–11]. Calcium antagonists (CA) were listed in textbooks as potential causes of GERD by relaxing the LES, thereby facilitating the reflux of gastric contents into the esophagus. Conversely, other studies have not found a significant association or have reported protective effects, creating ambiguity in the literature [12–14].

The relationship between amlodipine and GERD remains poorly understood, with conflicting evidence necessitating further investigation. Recent advancements in genetic epidemiology, particularly the application of Mendelian Randomization (MR), offer a promising approach to disentangle this complex relationship [15]. MR uses genetic variants as instrumental variables to assess causal relationships between exposures (such as drug use) and outcomes (such as disease risk), leveraging the random assortment of genes at conception to mitigate confounding factors and reverse causation [16]. This approach has been successfully employed in various studies to explore drug-disease associations and provide robust evidence for causal inferences.

Despite the potential of MR to clarify the effects of amlodipine on GERD, there is a paucity of research applying this method to investigate their relationship. Our study employs a bidirectional MR approach to elucidate the causal link between genetically predicted amlodipine use and GERD risk. By using publicly available genome-wide association study (GWAS) [17] datasets, we aim to identify genetic variants associated with amlodipine exposure and GERD outcomes, thereby providing insights into the directionality and magnitude of their association. Furthermore, the study explores the regulatory mechanisms underlying this association by constructing a competing endogenous RNA (ceRNA) network. The ceRNA hypothesis posits that long non-coding RNAs (lncRNAs), microRNAs (miRNAs), and mRNAs can regulate

each other's expression through shared miRNA response elements, creating a complex regulatory network [18–21]. By integrating data from multiple databases such as TargetScan, miRDB, miRanda, miRWalk, and miRcode, we aim to identify key RNA molecules involved in the ceRNA network and understand their interactions with amlodipine target genes.

Our study investigates the relationship between amlodipine, a widely used calcium channel blocker, and the risk of GERD. While traditional observational studies have yielded conflicting evidence regarding the impact of amlodipine on GERD, our research leverages a Mendelian Randomization (MR) approach to provide more robust and causally informative insights. Specifically, this study is distinguished by its bidirectional MR analysis, which not only examines the causal effect of amlodipine on GERD but also explores the potential reverse causality—whether a genetic predisposition to GERD could influence amlodipine use. Furthermore, our study integrates the construction of a competing endogenous RNA (ceRNA) network, a novel approach that allows us to explore the molecular mechanisms underlying the genetic associations observed in the MR analysis. By identifying key regulatory RNA molecules and pathways involved in GERD pathogenesis, this research offers new insights that could lead to targeted therapeutic interventions.

## Materials and methods

### Study design

The data utilized in our analysis were publicly accessible and had been authorized by the institutional review board in the relevant studies. Hence, additional sanctions were unnecessary. The article and its supplements contain all the results that were generated. We conducted a bidirectional Mendelian Randomization (MR) study to explore the relationship between amlodipine and the risk of gastroesophageal reflux disease (GERD). We searched drug target databases to identify targets associated with amlodipine. Based on the identified drug targets, we constructed a competing endogenous RNA (ceRNA) network. The identified drug targets underwent enrichment analysis using both Gene Ontology (GO) and KEGG pathway databases. The GeneMANIA network tool was used to visualize and further explore the functional network of the target genes. We downloaded GERD-related datasets from the Gene Expression Omnibus (GEO) database, and differential gene expression analysis was performed on the dataset to identify genes that are significantly associated with GERD. Finally, we intersected the results of drug target genes and differential analysis to further verify the previous analysis.

### Data sources and processing

The GWAS data used in our study were all publicly available, and the participants were of European ancestry. Data on amlodipine and GERD were drawn both from the Neale Lab consortium, which is available at https://gwas.mrcieu.ac.uk/datasets/ukb-a-150/ and https://gwas.mrcieu.ac.uk/datasets/ukb-a-70/,respectively(Treatment/medication code: amlodipine: including 13,693 cases and 323,466 controls, gastro-esophageal reflux including 14,316 cases and 322,843 controls). GEO data can be found here: GSE77563 (https://www.ncbi.nlm.nih.gov/geo/query/acc.cgi?acc=GSE77563). Drug target information can be found through the following websites: dgidb.genome.wustl.edu and DrugBank Online | Database for Drug and Drug Target Info.

### Instrumental variable selection and data harmonization

We included SNPs that were genome-wide significant ($P < 5 \times 10^{-8}$). These SNPs were then clustered based on linkage disequilibrium (window size = 10,000 kb and $r^2 < 0.001$). Estimated

levels of linkage disequilibrium from the 1000 Genomes Project based on European samples [22]. If a particular exposed SNP was not present in the outcome dataset, proxy SNPs were used by LD tagging. Palindromic and ambiguous SNPs were excluded from IVs for Mendelian randomization analysis [23]. For each instrument (SNP), we calculate the proportion of variance explained ($R^2$) for both the exposure and the outcome, $R^2 = 2 \times (\beta \text{exposure/outcome})$ $2 \times \text{eaf.exposure/outcome} \times (1 - \text{eaf.exposure/outcome})$. Compare the $R^2$ values for the exposure and the outcome. If the $R^2$ for the exposure was greater than the $R^2$ for the outcome, the instrument is considered valid. Remove instruments where the $R^2$ for the outcome is greater than the $R^2$ for the exposure, as these are considered weak instruments [24, 25].

## Sensitivity analysis

Heterogeneity between SNPs was assessed using Cochran's Q statistic and funnel plots [26, 27]. Horizontal pleiotropy was detected using the MR-Egger intercept [28] method and the MR- PRESSO [29] method. If outliers were detected, they were removed, and we re-evaluated the MR causal estimates. If heterogeneity remained high after removal, the stability of the results was assessed using a random effects model, which is less susceptible to weaker SNP exposure associations. Finally, leave- one-out analysis was used to validate the effect of each SNP on the overall causal estimates.

## Functional enrichment analysis of drug target genes

The target genes identified underwent enrichment analysis using both Gene Ontology (GO) and KEGG pathway databases. This analysis helps to determine the biological functions, cellular components, and biological processes that these genes are predominantly involved in, as well as the metabolic or signaling pathways they participate in.

## GeneMANIA network visualization of drug target genes

The GeneMANIA network tool was used to visualize and further explore the functional network of target genes [30]. This tool generates a graphical representation of the gene interactions, including direct and indirect relationships, providing insights into the genetic architecture underlying the diseases studied. The network was analyzed to identify additional genes associated with the core genes, potentially expanding the understanding of the genetic influences between amlodipine and gastroesophageal reflux disease.

## Construction of a ceRNA network of drug target genes and identification of core targets

Target mRNAs of differentially expressed miRNAs were predicted by an integration of TargetScan (targetscan. org [18], miRDB (http://mirdb.org [19] miRanda (miranda.org/) and miRWalk (http://mirwalk.umm.uni-heidelberg.de/) databases. Furthermore, miRNA-lncRNA relationships were predicted by implementing the miRcode database (http://www.mircode.org) [20]. The miRNA-mRNA and miRNA-lncRNA interactions were imported into Cytoscape software (version: 3.10.2) to construct a ceRNA network [21]. CytoHubba, a Cytoscape plugin (v0.1), was used to identify core targets within the ceRNA network. Using the Degree, MCC, and Closeness algorithms in CytoHubba, several nodes were identified as core targets within the ceRNA network.

## Differential expression analysis and verification of target genes

Differential expression analysis was performed using the limma package. The analysis focused on identifying genes with significant changes in expression between control and GERD

groups, defined by a log fold change (logFC) threshold of 0.585 and a P-value threshold of 0.05. A linear model was fitted to the data, followed by the application of contrasts to determine differential expression between the groups. Then, we intersected the differential expression results with the drug targets to further verify the targets of the relationship between drugs and diseases.

## Statistical analysis

We performed MR analysis using R software (version 4.3.2, http://www.r-project.org) and the "Two-Sample MR" package (version 0.5.11) [31]. MR-Pleiotropy RESidual Sum and Outlier (MR-PRESSO) and robust adjusted profile score (MR.RAPS) were performed using the R packages "MRPRESSO" and "MR. raps", respectively. And we applied a PhenoScanner search to assess all known phenotypes related to the considered genetic instruments in our analyses. The selected SNPs should not be associated with confounders of the exposure-outcome relationship. We minimized confounding by using SNPs in low linkage disequilibrium (LD, which referred to the non-random association of alleles at different loci. In other words, it described a situation where certain genetic variants (alleles) tended to be inherited together more often than would be expected by chance, due to their physical proximity on the same chromosome), and by applying sensitivity analyses, such as MR Egger and MR-PRESSO, to detect and correct for horizontal pleiotropy. The SNPs should affect the outcome (GERD risk) only through the exposure (amlodipine use) and not through alternative pathways. This assumption was critical to avoid bias in causal estimates. We addressed potential violations of this assumption using MR-PRESSO and by conducting a leave-one-out analysis to ensure robustness. Pleiotropy occurs when a single genetic variant influences multiple traits or phenotypes. In the context of MR, pleiotropy can be problematic if a genetic variant affects the outcome not only through the exposure of interest but also through other unrelated pathways. We used methods such as MR-Egger regression and MR-PRESSO to detect and correct for pleiotropy. These methods help ensure that our causal estimates are not biased by the pleiotropic effects of the genetic variants used as instruments. For all statistical tests, $P < 0.05$ was statistically significant.

## Results

### Association of coronary atherosclerosis with prostate cancer

After removing palindromic and ambiguous SNPs, there were 19 SNPs in amlodipine and 16 SNPs in gastroesophageal reflux disease as instrumental variables (**S1 and S2 Tables**). All five methods were used to estimate the causal relationship between genetically predicted amlodipine and GERD (**Table 1**). Across all five MR methods, there was broad and consistent support for the negative association of amlodipine with GERD (Inverse variance weighted odds ratio [OR] = 0.872 [95% CI, 0.812–0.937], P = 0.0002; Weighted median OR = 0.878 [95% CI, 0.798–0.966], P = 0.007). However, the results of our MR analysis showed no reverse causality for genetically predicted amlodipine with GERD (i.e., no causality for genetically predicted GERD and amlodipine). The OR was 1.005[95% CI, 0.903–1.119; p = 0.921] by using the IVW method. The results were all shown in **Table 1**.

### Sensitivity analysis

Several sensitivity analyses were used to examine and correct for the presence of pleiotropy in causal estimates. In our study, the MR-Egger intercept showed no evidence of pleiotropy (**Table 1**),and Cochran's Q-test and funnel plot showed no evidence of heterogeneity and

**Table 1. Mendelian randomization results and heterogeneity and pleiotropy test results of the correlation between amlodipine and gastroesophageal reflux disease.**

| exposure | outcome | method | nsnp | or | or_lci95 | or_uci95 | pval | heterogeneity-test | | | | pleiotropy-test | | |
|---|---|---|---|---|---|---|---|---|---|---|---|---|---|---|
| Amlodipine | gastro-oesophageal | MR Egger | 19 | 1.011 | 0.757 | 1.349 | 0.942 | | Q | Q_df | Q_pval | egger_intercept | se | pval |
| | | Weighted median | 19 | 0.878 | 0.798 | 0.966 | 0.007 | MR Egger | 11.221 | 17 | 0.845 | -0.0005 | 0.0005 | 0.316 |
| GWAS ID: ukb-a-150 | GWAS ID: ukb-a-70 | Inverse variance weighted | 19 | 0.872 | 0.812 | 0.937 | 0.0002 | Inverse variance weighted | 12.288 | 18 | 0.832 | | | |
| | | Simple mode | 19 | 0.827 | 0.700 | 0.977 | 0.038 | | | | | | | |
| | | Weighted mode | 19 | 0.857 | 0.742 | 0.991 | 0.052 | | | | | | | |
| gastro-oesophageal | Amlodipine | MR Egger | 16 | 1.074 | 0.719 | 1.605 | 0.732 | | | | | | | |
| | | Weighted median | 16 | 0.996 | 0.867 | 1.145 | 0.957 | MR Egger | 16.900 | 14 | 0.262 | -0.000204 | 0.0006 | 0.742 |
| GWAS ID: ukb-a-70 | GWAS ID: ukb-a-150 | Inverse variance weighted | 16 | 1.005 | 0.903 | 1.119 | 0.921 | Inverse variance weighted | 17.036 | 15 | 0.317 | | | |
| | | Simple mode | 16 | 0.997 | 0.755 | 1.315 | 0.982 | | | | | | | |
| | | Weighted mode | 16 | 0.997 | 0.755 | 1.316 | 0.982 | | | | | | | |

asymmetry between these SNPs in the causal relationship between these SNPs (**Table 1** and **S1 Fig**).The effect of each SNP on the overall causal estimates was verified by leave-one-out analysis (**S2 Fig**). After removing each SNP, we systematically performed the MR analysis again for the remaining SNPs. The results remained consistent, indicating that all SNPs were calculated to make the causal relationship significant.

## Functional enrichment analysis of drug target genes

By querying dgidb.genome.wustl.edu and DrugBank Online | Database for Drug and Drug Target Info., we obtained 19 amlodipine targets (*NPPA, CACNA1B, CACNB2, CACNA2D1, CACNA1C, CACNA1D, CACNG1, CACNA1S, ACE, AGT, CYP3A5, CYP3A4, SMPD1, CA1, CACNA2D3, CACNB1, CACNA1B, CACNA1I, CACNA1C*). After identifying the target genes of amlodipine we next explored the mechanisms underlying these genes' functions using the GeneMANIA database (http://genemania.org/), an online platform used to search for proteins associated with specific genes and gene sets [30]. Gene Ontology (GO) enrichment was conducted to classify the genes into Biological Process (BP), Cellular Component (CC), and Molecular Function (MF) categories using the clusterProfiler package in R. KEGG pathway enrichment analysis was performed to identify the metabolic and signaling pathways in which these genes are involved. We identified the potential mechanisms of these 19 genes. We found that these genes were enriched in biological activities, such as calcium ion transport, calcium ion transmembrane transport via high voltage-gated calcium channel, calcium ion transmembrane transport, calcium signaling pathway, chemical carcinogenesis—receptor activation and cortisol synthesis and secretion (**Fig 1A–1C**).

## Construction of a ceRNA network in drug target genes and Identification of core targets

After matching the miRNA-mRNA and lncRNA-miRNA pairs, a ceRNA network was established for target genes (**Fig 2A**). The net- work was comprised of 218 lncRNAs, 57 miRNAs, and 14 mRNAs (CACNA1C, CACNA1D, CACNB1, ACE, CACNB2, CACNA1B,

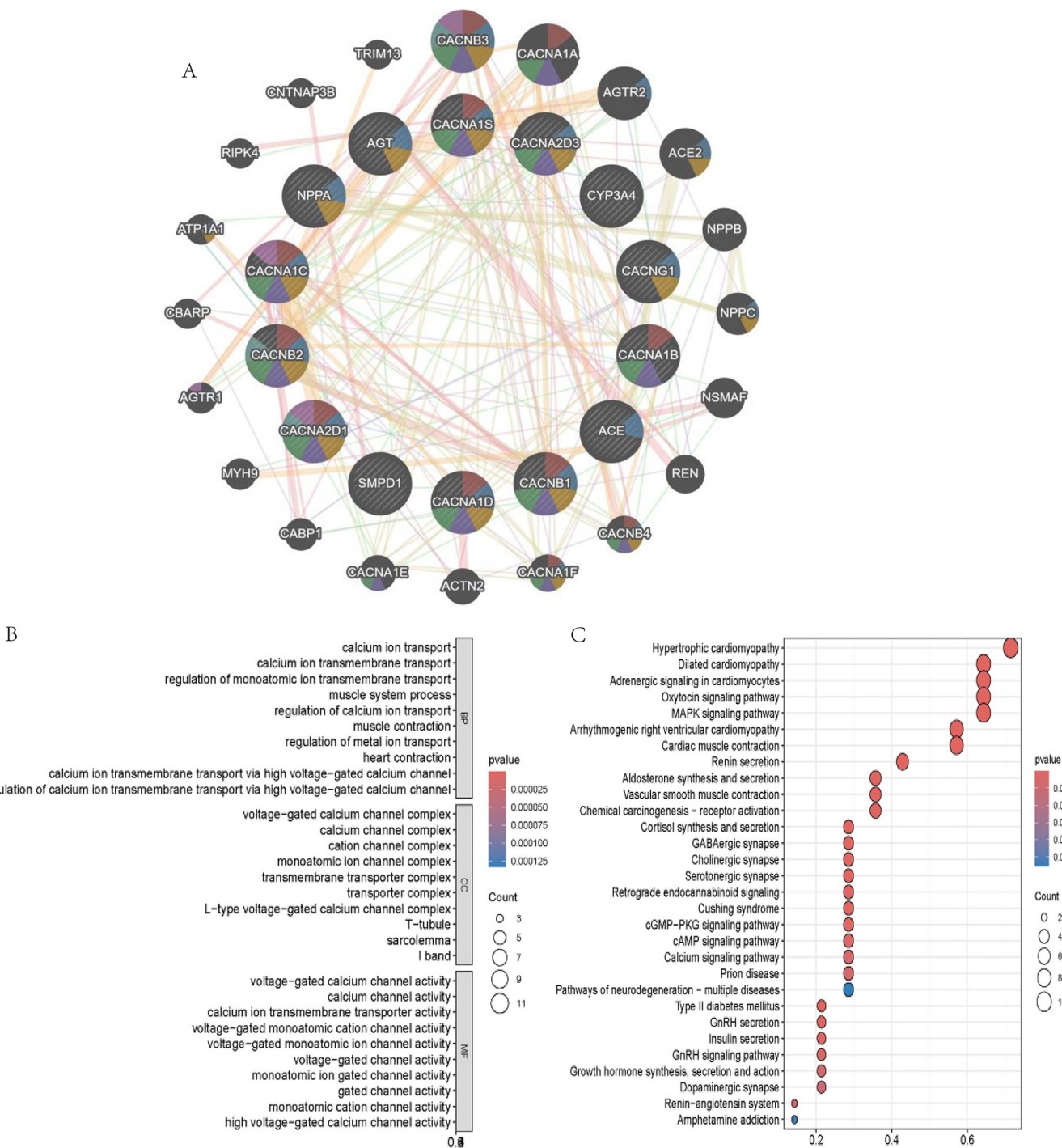

**Fig 1. GO/ KEGG enrichment and GeneMANIA network visualization of drug target genes.** (A) GeneMANIA Network of Drug Target Genes. Each node represents a gene, while the edges (lines connecting the nodes) indicate predicted or known interactions between these genes. The size of each node correlates with the gene's degree of connectivity within the network, highlighting the genes with the most interactions. Prominent genes such as *CACNB2*, *CACNA1C*, and *CACNA2D1* are highlighted due to their central roles in calcium ion transport and signaling pathways. The colored segments within each node represent the various types of interactions or functions associated with the gene, such as co-expression, co-localization, and physical interactions. The multicolored edges indicate different types of relationships between the genes, reflecting the complex interplay of signaling pathways involved. (B) The bubble plot of Gene Ontology (GO) enrichment analysis. The plot highlights several critical biological processes such as *regulation of calcium ion transmembrane transport via high voltage-gated calcium channel* and *muscle contraction*; The analysis also emphasizes cellular components like *the L-type voltage-gated calcium channel complex* and *T-tubule*, which are integral to calcium signaling in muscle cells; key molecular functions identified include *calcium ion transmembrane transporter activity*

and *voltage-gated calcium channel activity* (C) The bubble plot of KEGG enrichment analysis. The plot highlights several significant pathways, including *Calcium signaling pathway*, *Vascular smooth muscle contraction*, and *Renin-angiotensin system*. The enrichment of pathways such as the *Calcium signaling pathway* and *Vascular smooth muscle contraction* supports the hypothesis that amlodipine's effects on GERD may be mediated through its influence on calcium ion transport and smooth muscle function. Additionally, the involvement of pathways like *Renin-angiotensin system* highlights the potential systemic effects of amlodipine on cardiovascular health, which could indirectly influence GERD risk.

CACNA2D3, CACNG1, SMPD1, CACNA2D1, AGT, CACNA1S, CYP3A4, NPPA). A comparative analysis of the results from different algorithms showed some overlap, with hsa-miR-149-3p, hsa-miR-650, hsa-miR-612, hsa-miR-1976, CACNA1C, hsa-miR-515-5p, hsa-miR-129-5p, hsa-miR-541-3p, CACNB2, appearing in the top lists of three algorithms (**Fig 2B, Table 2**). This overlap suggests that these nodes are likely to be crucial regulators in the ceRNA network.

## Differential expression analysis and verification of target genes

The mRNA expression profiles of 21 gastroesophageal reflux disease and 19 normal samples were extracted from the GSE77563 dataset. Compared with normal samples, there were 353 up-regulated and 114 down-regulated mRNAs in the gastroesophageal reflux disease samples (**Fig 3A and 3B**). Then, we intersected the differential expression results with the drug targets and found that *CACNB2* was the only intersection gene (**Fig 3C**).

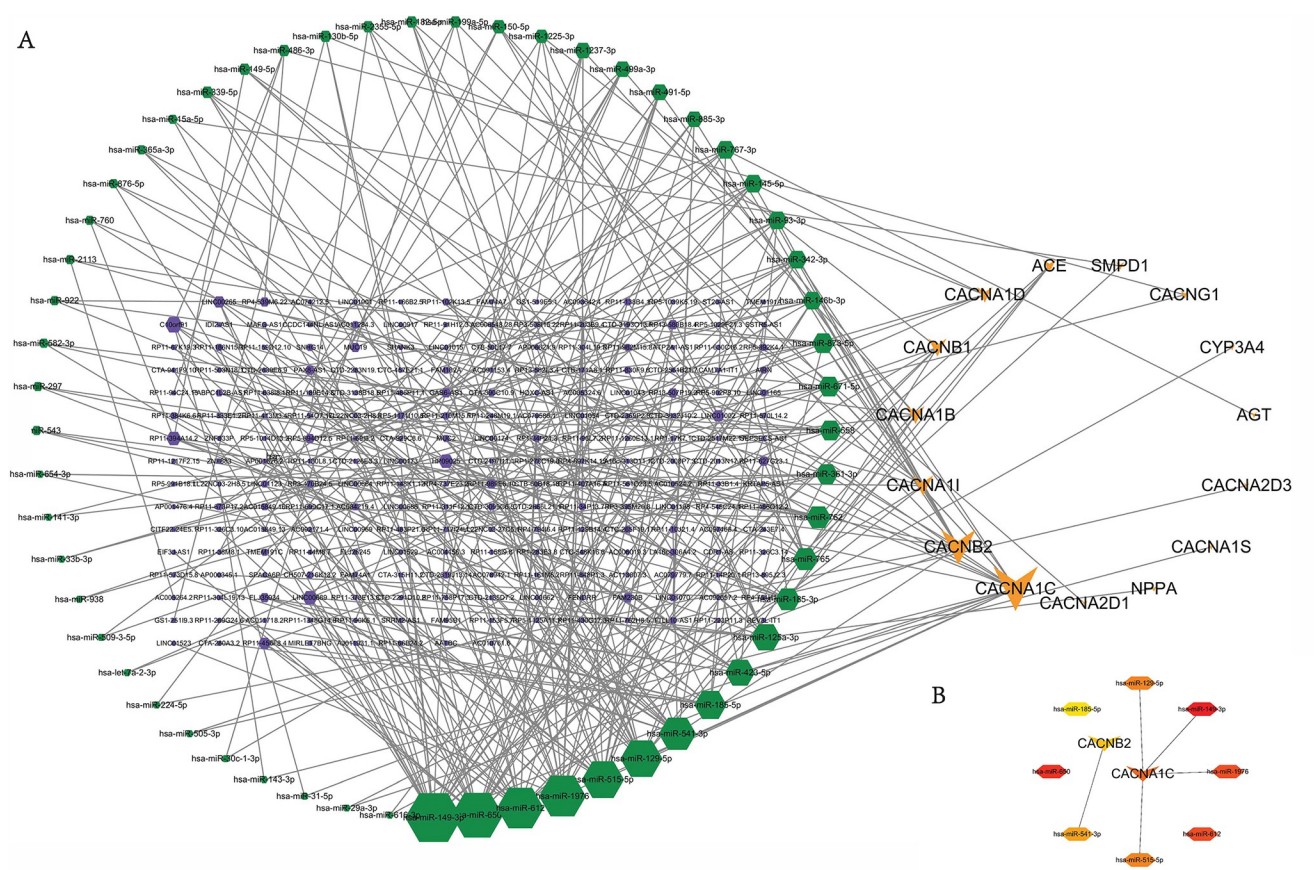

**Fig 2. Establishment of a ceRNA network based on the targets for amlodipine and a core targets network based on ceRNA network. (A)** ceRNA network based on the targets for amlodipine. yellow vs. represent drug targets, green hexagons represent the miRNAs, and purple hexagons represent the lncRNAs. The larger of the size, the larger the degree. **(B)** The core targets network based on ceRNA network. The darker the color, the larger the degree.

**Table 2. The characteristics of nodes in the ceRNA network.**

| node name | MCC | Degree | Closeness |
|---|---|---|---|
| hsa-miR-149-3p | 22 | 22 | 103.2488 |
| hsa-miR-650 | 20 | 20 | 89.59048 |
| hsa-miR-612 | 18 | 18 | 83.89762 |
| hsa-miR-1976 | 18 | 18 | 91.56548 |
| CACNA1C | 17 | 17 | 110.0012 |
| hsa-miR-515-5p | 16 | 16 | 90.77262 |
| hsa-miR-129-5p | 16 | 16 | 89.65357 |
| hsa-miR-541-3p | 14 | 14 | 88.52024 |
| CACNB2 | 13 | 13 | 91.50119 |
| hsa-miR-185-5p | 12 | 12 | 80.04286 |
| hsa-miR-423-5p | 11 | 11 | 90.29881 |
| hsa-miR-125a-3p | 11 | 11 | 77.79167 |
| hsa-miR-765 | 9 | 9 | 83.35952 |
| hsa-miR-762 | 9 | 9 | 77.34167 |
| hsa-miR-185-3p | 9 | 9 | 82.81548 |
| hsa-miR-873-5p | 8 | 8 | 83.43095 |
| hsa-miR-671-5p | 8 | 8 | 91.78333 |
| hsa-miR-558 | 8 | 8 | 77.19286 |
| hsa-miR-361-3p | 8 | 8 | 81.98452 |
| CACNA1I | 7 | 7 | 83.03452 |
| hsa-miR-145-5p | 7 | 7 | 76.94405 |
| hsa-miR-146b-3p | 7 | 7 | 77.28095 |
| hsa-miR-342-3p | 7 | 7 | 81.50595 |
| hsa-miR-767-3p | 7 | 7 | 75.80357 |
| hsa-miR-93-3p | 7 | 7 | 73.30119 |
| CACNB1 | 6 | 6 | 85.57976 |
| CACNA1B | 6 | 6 | 78.04921 |
| hsa-miR-499a-3p | 6 | 6 | 66.29892 |
| hsa-miR-1237-3p | 6 | 6 | 62.64654 |
| hsa-miR-885-3p | 6 | 6 | 82.625 |

## Discussion

The study's main findings highlight a significant negative association between genetically predicted amlodipine use and the risk of gastroesophageal reflux disease (GERD). Using a bidirectional Mendelian Randomization (MR) approach, the analysis revealed that amlodipine, a commonly prescribed antihypertensive medication, is inversely associated with the occurrence of GERD. This finding was consistent across various MR methods, with the inverse variance weighted (IVW) method showing an odds ratio (OR) of 0.872 (95% CI, 0.812–0.937, P = 0.0002) and the weighted median method indicating an OR of 0.878 (95% CI, 0.798–0.966, P = 0.007).The sensitivity analyses further supported the robustness of these results, demonstrating no evidence of pleiotropy or heterogeneity among the single nucleotide polymorphisms (SNPs) used as instrumental variables. Additionally, the leave-one-out analysis confirmed that the findings were not driven by any single SNP. The study's findings clearly demonstrate a lack of reverse causality between genetically predicted gastroesophageal reflux disease (GERD) and the use of amlodipine. Using a bidirectional Mendelian Randomization (MR) approach, the analysis specifically investigated whether a genetic predisposition to

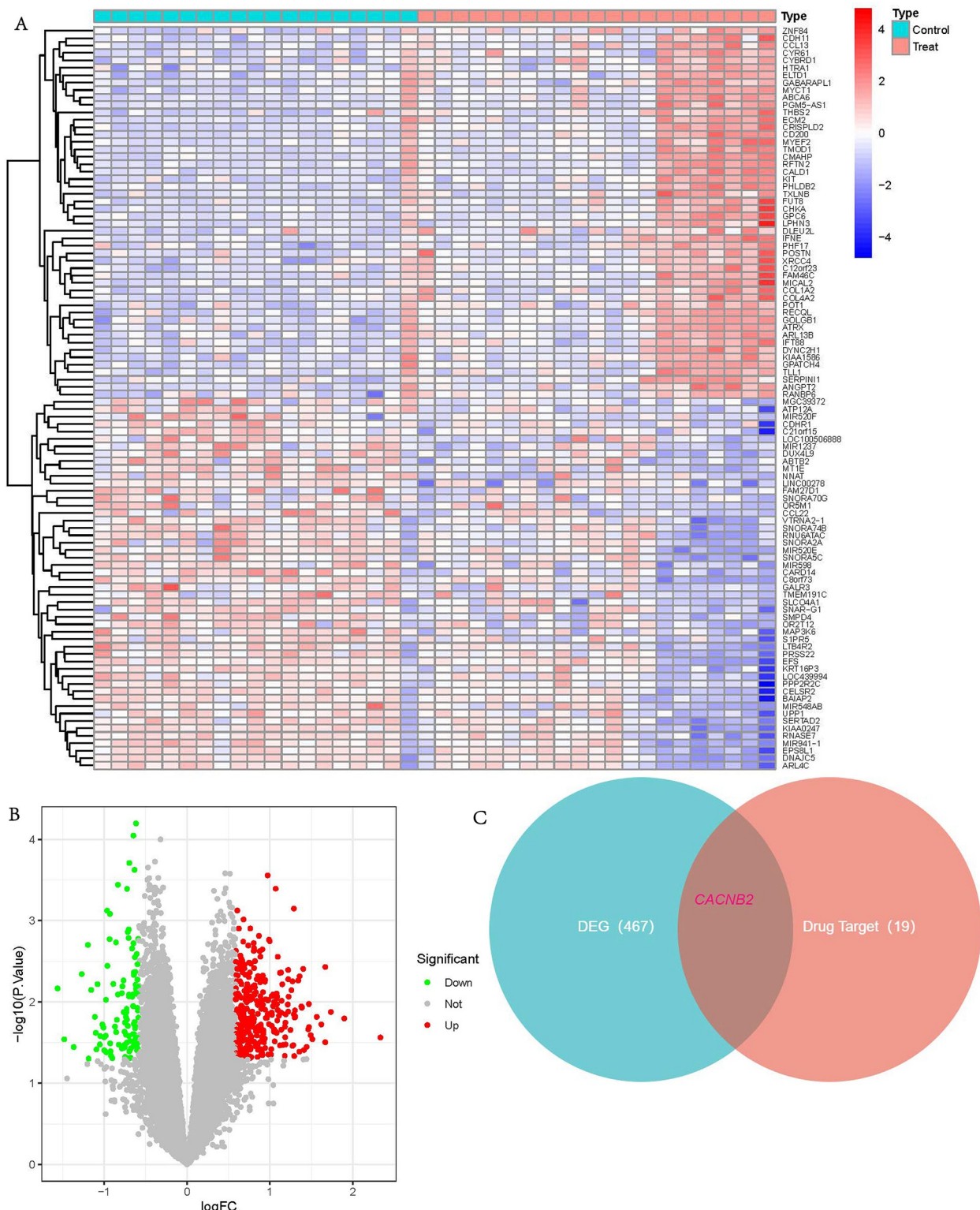

**Fig 3.** Differential Expression Analysis and Verification of Target Genes (**A, B**) heat and volcano map for differential expression genes between gastroesophageal reflux disease and normal specimens. green: down-regulation; red: up-regulation; grey: not significant (**C**) Venn diagrams for the common gene in the differential expression genes and drug target genes.

GERD might influence the likelihood of amlodipine use. The results showed no significant causal effect of genetically predicted GERD on amlodipine use. This was evidenced by the inverse variance weighted (IVW) method, which provided an odds ratio (OR) of 1.005 (95% CI, 0.903–1.119, P = 0.921). This OR, close to 1, indicates no association between GERD and the use of amlodipine. Additionally, sensitivity analyses, including the MR-Egger intercept and Cochran's Q-test, showed no evidence of pleiotropy or heterogeneity, further supporting the robustness of these findings. Therefore, while the study found a negative association between genetically predicted amlodipine use and GERD risk, it did not find any evidence to suggest that a genetic predisposition to GERD influences the use of amlodipine. This lack of reverse causality strengthens the conclusion that amlodipine may have a protective effect against GERD without being affected by the presence of GERD itself.

Amlodipine, a calcium channel blocker, is widely used for the treatment of hypertension and angina [7, 8]. It is generally accepted from the textbook that calcium channel blockers may relax the lower esophageal sphincter, which may lead to gastroesophageal reflux disease symptoms in susceptible individuals. However, the findings of this study provide genetic evidence countering this clinical observation, suggesting that amlodipine might reduce the risk of GERD. This contrasts with the initial assumptions and highlights the complexity of drug-disease interactions. The study's enrichment analysis identified several biological processes and pathways associated with amlodipine targets, including calcium ion transport and the calcium signaling pathway. These pathways are crucial in the regulation of smooth muscle tone, including the lower esophageal sphincter [32–34]. The protective effect observed in the study might be linked to amlodipine's broader pharmacological impacts beyond simple muscle relaxation, potentially involving anti-inflammatory and other systemic effects that reduce GERD risk [35–38]. The current study adds valuable genetic evidence specific to amlodipine, suggesting it may have a unique protective role against GERD, differing from other antihypertensive classes.

Previous Mendelian Randomization (MR) studies have explored the genetic determinants of GERD, but few have focused on specific drug effects like amlodipine [39–42]. This study employs a bidirectional MR approach to specifically examine the causal relationship between amlodipine and GERD. The study's results suggest that amlodipine might be a safer choice for hypertensive patients who are also at risk for GERD, contrasting with concerns over its potential to exacerbate GERD symptoms. This aligns with some clinical guidelines suggesting careful selection of antihypertensive medications in patients with concurrent GERD [43], emphasizing the importance of considering individual drug effects.

The Gene Ontology (GO) enrichment analysis classified the identified amlodipine target genes into several key biological processes. Notably, the targets were enriched in processes related to calcium ion transport and calcium ion transmembrane transport via high voltage-gated calcium channels. This finding is consistent with the primary pharmacological action of amlodipine, which involves blocking calcium channels to reduce calcium influx in vascular smooth muscle cells, leading to vasodilation and lowered blood pressure [7, 8, 44]. The analysis also identified significant enrichment in cellular components associated with the plasma membrane and voltage-gated calcium channel complexes. These cellular components are directly related to the sites where amlodipine exerts its effects, confirming the drug's targeted action on calcium channels located in the plasma membrane of muscle cells [45]. Molecular function analysis revealed that the target genes are involved in calcium channel activity and voltage-gated calcium channel activity. This aligns with amlodipine's mechanism as a calcium channel blocker, specifically targeting the L-type calcium channels (CACNA1C, CACNA1D, etc.) to inhibit calcium ion entry into cells, thereby exerting its therapeutic effects [46]. The KEGG pathway analysis identified several enriched pathways, including the calcium signaling

pathway, chemical carcinogenesis—receptor activation, and cortisol synthesis and secretion. The enrichment in the calcium signaling pathway is particularly significant, as it underscores the fundamental role of calcium ions in cellular signaling and muscle contraction. Amlodipine's modulation of this pathway is crucial for its therapeutic effects in treating hypertension and angina. The identification of pathways related to chemical carcinogenesis and cortisol synthesis suggests potential broader implications of amlodipine's effects. These findings might indicate additional roles for calcium signaling in these processes, although further research is needed to clarify these connections. The GeneMANIA network tool provided a graphical representation of the interactions between the target genes, highlighting both direct and indirect relationships. The visualization of this network helps to understand the genetic architecture underlying the effects of amlodipine. The network analysis identified additional genes associated with the core targets, potentially expanding the understanding of how amlodipine influences various biological processes and disease states. The construction of the ceRNA network further illuminated the regulatory interactions involving miRNAs, lncRNAs, and mRNAs. Core targets such as CACNA1C, CACNA1D, and CACNB2 were identified within this network, highlighting their central role in mediating the effects of amlodipine. The overlap of key nodes across different algorithms suggests that these genes and their regulatory networks are crucial in the context of amlodipine's action. In summary, the enrichment analysis results are consistent with the known biological functions and pathways involving amlodipine targets. The findings reinforce the understanding of amlodipine's role in modulating calcium signaling and provide additional insights into its broader biological impacts.

The identification of CACNB2 as a core target gene provides new insights into the genetic and molecular mechanisms underlying GERD. CACNB2 encodes the beta-2 subunit of voltage-gated calcium channels, which are crucial for regulating calcium ion influx in various tissues [47, 48]. Understanding its involvement offers a fresh perspective on how calcium signaling might influence GERD, potentially linking it to smooth muscle function and esophageal motility. Traditionally, GERD research has focused on factors like gastric acid secretion and lower esophageal sphincter (LES) relaxation [49, 50]. The ceRNA network analysis in our study highlighted key regulatory interactions involving calcium channel subunits, particularly CACNB2. This gene encodes the beta-2 subunit of voltage-gated calcium channels, which are critical for proper calcium ion transport and signaling [47]. Variations in CACNB2 expression or function could influence LES tone and esophageal motility, potentially contributing to the protective effect of amlodipine observed in our study. By stabilizing the function of calcium channels through its action on CACNB2 and related pathways, amlodipine may help maintain the integrity of the LES and prevent GERD symptoms. Calcium signaling is integral to the coordinated contraction and relaxation of gastrointestinal smooth muscles, which are essential for proper motility and function. By highlighting a gene involved in calcium signaling, this study broadens the scope of GERD research, encouraging investigations into how calcium channel regulation might affect esophageal physiology and GERD development. The findings open up interdisciplinary research opportunities, combining gastroenterology, genetics, and pharmacology. Exploring how genetic variations in calcium channel subunits affect GERD can lead to a more integrated understanding of the disease, fostering collaborations across these fields. Calcium ions play a vital role in muscle contraction, including the smooth muscle of the esophagus and LES. The beta-2 subunit of the voltage-gated calcium channel (CACNB2) helps modulate this calcium influx. Variations in CACNB2 expression or function could affect the contractility and tone of the esophageal muscles and LES, potentially leading to the dysfunctions observed in GERD.A dysregulated LES is a hallmark of GERD, allowing gastric contents to reflux into the esophagus. If CACNB2 influences LES muscle tone, its dysregulation could contribute to inappropriate relaxation of the LES, thereby facilitating reflux episodes.

Understanding this connection could pinpoint CACNB2 as a therapeutic target for modulating LES function in GERD patients. The role of CACNB2 in calcium signaling suggests that calcium channel blockers, like amlodipine, might influence GERD symptoms through their effects on this pathway. This study's finding of a protective effect of amlodipine against GERD could be partly explained by its impact on CACNB2-mediated calcium signaling. Targeting CACNB2 directly or indirectly might offer new therapeutic strategies for GERD management. CACNB2 could serve as a biomarker for GERD susceptibility or treatment response. Genetic screening for variations in CACNB2 might help identify individuals at higher risk for GERD or those who might benefit most from calcium channel blockers or other treatments targeting calcium signaling pathways. The study suggests that amlodipine, a drug primarily used to treat hypertension and angina, might have a protective effect against gastroesophageal reflux disease (GERD). This opens the possibility of repurposing amlodipine for GERD treatment, providing an additional therapeutic option for patients suffering from this condition.

While the study provides valuable insights into the relationship between amlodipine and gastroesophageal reflux disease (GERD), several limitations must be acknowledged: The study relies on publicly available datasets, which may vary in quality and consistency. The data collection methods, sample sizes, and population characteristics might differ across datasets, potentially introducing biases or inconsistencies that could affect the results. While the use of proxy SNPs is necessary in certain instances, it is important to recognize and account for the potential biases and limitations they introduce. Future studies with access to more comprehensive genetic datasets may help to address these limitations and further validate the causal relationship between amlodipine use and GERD risk. Using publicly available datasets means the researchers have no control over the data collection process. This could limit the ability to address specific research questions or to adjust for potential confounders comprehensively. Public datasets might not include all the necessary variables or have incomplete data, which could limit the depth of the analysis. For example, detailed clinical information on GERD severity, treatment history, and other comorbidities might be lacking.

The study's findings are based on bioinformatics and statistical analyses without experimental validation. Functional studies, such as laboratory experiments and clinical trials, are necessary to confirm the biological relevance of the identified genetic interactions and pathways.

## Conclusion

This study provides insights into the genetic relationship between amlodipine use and gastroesophageal reflux disease (GERD) through a comprehensive bidirectional Mendelian Randomization (MR) approach. The findings reveal a protective effect of genetically predicted amlodipine use against GERD, supported by robust sensitivity analyses that confirm the absence of pleiotropy and heterogeneity. Notably, the study identifies CACNB2 as a core target gene, highlighting its potential role in the pathophysiology of GERD and suggesting new therapeutic avenues.

## Supporting information

**S1 Checklist. STROBE-MR checklist of recommended items to address in reports of Mendelian randomization studies[1] [2].**
(DOCX)

**S1 Table. Characteristics of significant SNPs with genome-wide associations (P<5×10−8) for amlodipine on gastroesophageal reflux disease (GERD).**
(XLSX)

**S2 Table. Characteristics of significant SNPs with genome-wide associations (P<5×10−8) for gastroesophageal reflux disease (GERD) on amlodipine.**
(XLSX)

**S1 Fig. Funnel plots to visualize overall heterogeneity of MR estimates for the effect of amlodipine on gastroesophageal reflux disease (GERD).** This funnel plot visualizes the relationship between the effect size (βIV) and the standard error (SEIV) of the instrumental variables (IVs) used in the Mendelian Randomization (MR) analysis of the relationship between amlodipine use and GERD risk. The lack of asymmetry in this plot suggests minimal bias due to heterogeneity among the SNPs. This supports the reliability of the causal estimates obtained from the MR analysis, indicating a robust association between genetically predicted amlodipine use and GERD risk.
(XLSX)

**S2 Fig. Leave-out plot to visualize causal effect of amlodipine on gastroesophageal reflux disease (GERD).** The x-axis represents the estimated effect size (beta coefficient) of amlodipine on GERD, while each point on the y-axis corresponds to the exclusion of a particular SNP. The "All" line represents the MR estimate when all SNPs are included, while each subsequent line shows the MR estimate when one specific SNP is removed from the analysis. The consistency of the effect size estimates across all leave-one-out iterations indicates that no single SNP is driving the overall association between amlodipine use and GERD risk. This robustness strengthens the confidence in the MR findings, suggesting that the observed protective effect of amlodipine on GERD is not the result of outlier SNPs or potential pleiotropy.
(XLSX)

## Acknowledgments

Summary statistics for the genetic associations with coronary atherosclerosis and prostate cancer were obtained from GWAS and GEO database, and we thank all investigators for sharing the genome-wide summary statistics.

## Author Contributions

**Conceptualization:** Liuzhao Zhang, Quanwang Chu, Shuyue Jiang.

**Formal analysis:** Bo Shao.

**Methodology:** Liuzhao Zhang.

**Project administration:** Liuzhao Zhang.

**Supervision:** Quanwang Chu.

**Writing – original draft:** Liuzhao Zhang, Quanwang Chu, Shuyue Jiang, Bo Shao.

**Writing – review & editing:** Liuzhao Zhang, Quanwang Chu, Shuyue Jiang, Bo Shao.

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
