## [Decision Letter · Decision Letter 0]

26 Jul 2024

PONE-D-24-22095Genetic Evidence for Amlodipine's Protective Role in Gastroesophageal Reflux Disease: A Focus on CACNB2PLOS ONE

Dear Dr. Shao,

Thank you for submitting your manuscript to PLOS ONE. After careful consideration, we feel that it has merit but does not fully meet PLOS ONE’s publication criteria as it currently stands. Therefore, we invite you to submit a revised version of the manuscript that addresses the points raised during the review process.

We look forward to receiving your revised manuscript.

Kind regards,

Asif Jan

Academic Editor

PLOS ONE

Journal Requirements:

2. Please remove your figures from within your manuscript file, leaving only the individual TIFF/EPS image files, uploaded separately. These will be automatically included in the reviewers’ PDF.

3. We notice that your supplementary figures are uploaded with the file type 'Other'. Please amend the file type to 'Supporting Information'. Please ensure that each Supporting Information file has a legend listed in the manuscript after the references list.

Additional Editor Comments:

Please incorporate reviewers comments and re-submit.

Reviewer 1.

Overall the “Genetic Evidence for Amlodipine's Protective Role in Gastroesophageal Reflux Disease: A Focus on CACNB2” is a very interesting research manuscript. Very much informative and well written, however needs some minor corrections.

1) The overall English is easy and understandable, however needs improvement.

2) Introduction chapter is relevant, however needs some latest inclusion of some references relevant to the context.

3) The methodology is relevant and very well written.

4) The results chapter is also written in a very good way.

5) Remove double spacing in Study Design line 3.

6) Check the last paragraph in Study Design for grammatical mistake.

7) Correct the spacing errors in Data sources and processing section.

8) Correct the grammatical mistake and spacing errors in section Instrumental variable selection and data harmonization.

9) No need to write Version as capital in section Construction of ceRNA network of Drug Target Genes and Identification of Core Targets.

10) Too many spacing errors. Kindly correct all the spacing errors in the overall research manuscript.

I accept the manuscript after the minor correction.

Reviewer 2:

This is a well written mansucript which investigates the relationship between genetically predicted amlodipine use and the risk of gastroesophageal reflux disease (GERD) using a bidirectional Mendelian Randomization approach, finding a significant protective effect of amlodipine against GERD, supported by genetic and molecular analyses.

I would siggest some minor corrections before considering it for publication.

In mAterial and methods;

Materials and Methods, improve clarity and conciseness of the following statement.

"The data utilized in our analysis were openly accessible and had been authorized by the institutional review board in the relevant studies."

Suggested revision: "The data used in our analysis were publicly available and had been authorized by the institutional review board in the relevant studies."

Overall, after making this minor change, this study can be published in its current form.

Reviewer 3:

Major Concerns

1. Methodological Clarity and Justification:

o Instrumental Variable Selection: The paper describes the selection and clustering of SNPs but lacks detail on the criteria used to validate these SNPs as strong instruments. While the linkage disequilibrium and significance thresholds are mentioned, the justification for excluding certain SNPs is not clearly explained.

o Data Harmonization: The harmonization process, especially handling palindromic and ambiguous SNPs, needs more clarity. The rationale behind proxy SNP usage and its impact on the results should be more thoroughly discussed.

2. Results Interpretation:

o Association vs. Causation: The paper concludes a protective effect of amlodipine on GERD risk. However, the distinction between association and causation needs stronger emphasis, especially given the nature of MR studies.

3. Functional Analysis and Biological Pathways:

o Enrichment Analysis: The connection between identified pathways and GERD is inferred but not deeply explored. How calcium ion transport and related pathways specifically influence GERD pathophysiology should be more explicitly discussed so that readers may better understand it.

4. General

o Comparison with Other Medications: While amlodipine is highlighted, comparisons with other calcium channel blockers or antihypertensive medications are not well developed. This comparison could strengthen the argument for amlodipine's unique role.

Minor Concerns

1. Writing and Presentation:

o Abstract and Introduction: The abstract is comprehensive, but the introduction could benefit from a clearer statement of the study's novelty and specific hypotheses.

o Figures and Tables: The figures (e.g., funnel plots, enrichment analyses) are informative, but the legends should provide more contexts to aid interpretation.

2. Statistical Methods:

o Software and Packages: While the use of R and specific MR packages is mentioned, a more detailed description of the statistical methods and assumptions would help readers replicate the study.

o Sensitivity Analyses: The results of sensitivity analyses are presented, but the implications of findings like the MR-Egger intercept and Cochran’s Q-test are not explained.

3. Technical Accuracy:

o Genetic Terms and Definitions: Some genetic terms (e.g., linkage disequilibrium, pleiotropy) are used without definitions. Including brief explanations would benefit readers who are not geneticists.

o Gene Lists and Databases: The sources of gene lists and databases used for functional analysis are cited, but the process of selecting and validating these resources should be clearer.

Decision: Minor revision

Reviewers' comments:

Reviewer's Responses to Questions

**Comments to the Author**

1. Is the manuscript technically sound, and do the data support the conclusions?

Reviewer #1: Yes

Reviewer #2: Yes

Reviewer #3: Yes

2. Has the statistical analysis been performed appropriately and rigorously? 

Reviewer #1: Yes

Reviewer #2: Yes

Reviewer #3: Yes

3. Have the authors made all data underlying the findings in their manuscript fully available?

Reviewer #1: Yes

Reviewer #2: Yes

Reviewer #3: Yes

4. Is the manuscript presented in an intelligible fashion and written in standard English?

Reviewer #1: Yes

Reviewer #2: Yes

Reviewer #3: Yes

5. Review Comments to the Author

Reviewer #1: REVIEW REPORT OF PONE-D-24-22095 - [EMID: dce6b8afba788c30]

Overall the “Genetic Evidence for Amlodipine's Protective Role in Gastroesophageal Reflux Disease: A Focus on CACNB2” is a very interesting research manuscript. Very much informative and well written, however needs some minor corrections.

1) The overall English is easy and understandable, however needs improvement.

2) Introduction chapter is relevant, however needs some latest inclusion of some references relevant to the context.

3) The methodology is relevant and very well written.

4) The results chapter is also written in a very good way.

5) Remove double spacing in Study Design line 3.

6) Check the last paragraph in Study Design for grammatical mistake.

7) Correct the spacing errors in Data sources and processing section.

8) Correct the grammatical mistake and spacing errors in section Instrumental variable selection and data harmonization.

9) No need to write Version as capital in section Construction of ceRNA network of Drug Target Genes and Identification of Core Targets.

10) Too many spacing errors. Kindly correct all the spacing errors in the overall research manuscript.

I accept the manuscript after the minor correction.

Reviewer #2: This is a well written mansucript which investigates the relationship between genetically predicted amlodipine use and the risk of gastroesophageal reflux disease (GERD) using a bidirectional Mendelian Randomization approach, finding a significant protective effect of amlodipine against GERD, supported by genetic and molecular analyses.

I would siggest some minor corrections before considering it for publication.

In mAterial and methods;

Materials and Methods, improve clarity and conciseness of the following statement.

"The data utilized in our analysis were openly accessible and had been authorized by the institutional review board in the relevant studies."

Suggested revision: "The data used in our analysis were publicly available and had been authorized by the institutional review board in the relevant studies."

Overall, after making this minor change, this study can be published in its current form.

Reviewer #3: Major Concerns

1. Methodological Clarity and Justification:

o Instrumental Variable Selection: The paper describes the selection and clustering of SNPs but lacks detail on the criteria used to validate these SNPs as strong instruments. While the linkage disequilibrium and significance thresholds are mentioned, the justification for excluding certain SNPs is not clearly explained.

o Data Harmonization: The harmonization process, especially handling palindromic and ambiguous SNPs, needs more clarity. The rationale behind proxy SNP usage and its impact on the results should be more thoroughly discussed.

2. Results Interpretation:

o Association vs. Causation: The paper concludes a protective effect of amlodipine on GERD risk. However, the distinction between association and causation needs stronger emphasis, especially given the nature of MR studies.

3. Functional Analysis and Biological Pathways:

o Enrichment Analysis: The connection between identified pathways and GERD is inferred but not deeply explored. How calcium ion transport and related pathways specifically influence GERD pathophysiology should be more explicitly discussed so that readers may better understand it.

4. General

o Comparison with Other Medications: While amlodipine is highlighted, comparisons with other calcium channel blockers or antihypertensive medications are not well developed. This comparison could strengthen the argument for amlodipine's unique role.

Minor Concerns

1. Writing and Presentation:

o Abstract and Introduction: The abstract is comprehensive, but the introduction could benefit from a clearer statement of the study's novelty and specific hypotheses.

o Figures and Tables: The figures (e.g., funnel plots, enrichment analyses) are informative, but the legends should provide more contexts to aid interpretation.

2. Statistical Methods:

o Software and Packages: While the use of R and specific MR packages is mentioned, a more detailed description of the statistical methods and assumptions would help readers replicate the study.

o Sensitivity Analyses: The results of sensitivity analyses are presented, but the implications of findings like the MR-Egger intercept and Cochran’s Q-test are not explained.

3. Technical Accuracy:

o Genetic Terms and Definitions: Some genetic terms (e.g., linkage disequilibrium, pleiotropy) are used without definitions. Including brief explanations would benefit readers who are not geneticists.

o Gene Lists and Databases: The sources of gene lists and databases used for functional analysis are cited, but the process of selecting and validating these resources should be clearer.

Decision: Minor revision

6. PLOS authors have the option to publish the peer review history of their article (what does this mean?). If published, this will include your full peer review and any attached files.

Reviewer #1: **Yes: **Dr. Naveed Rahman

Reviewer #2: **Yes: **Tahir Muhammad

Reviewer #3: **Yes: **Dr. Waheed Ali Shah

---

## [Author Response · Author response to Decision Letter 0]

9 Aug 2024

Major Concerns

1. Methodological Clarity and Justification:

o Instrumental Variable Selection: The paper describes the selection and clustering of SNPs but lacks detail on the criteria used to validate these SNPs as strong instruments. While the linkage disequilibrium and significance thresholds are mentioned, the justification for excluding certain SNPs is not clearly explained.

o Response: Thank you for your insightful comment regarding the selection and clustering of SNPs. We agree that further clarification is needed. In our study, we selected SNPs based on genome-wide significance (P < 5 × 10−8) and ensured independence by clustering them with a linkage disequilibrium threshold of r² < 0.001 within a 10,000 kb window. This approach was chosen to minimize confounding due to linkage disequilibrium. SNPs were excluded if they did not meet these criteria or if they were identified as palindromic and ambiguous, as these can introduce biases in Mendelian Randomization (MR) analysis. We will revise the manuscript to provide a more detailed explanation of these criteria and their implementation. For example, we applied a PhenoScanner search to assess all known phenotypes related to the considered genetic instruments in our analyses.

o Data Harmonization: The harmonization process, especially handling palindromic and ambiguous SNPs, needs more clarity. The rationale behind proxy SNP usage and its impact on the results should be more thoroughly discussed.

o Response: We appreciate your concern regarding data harmonization. In our study, palindromic SNPs were carefully managed to avoid strand ambiguity, and ambiguous SNPs were excluded from the analysis to maintain data integrity. When necessary, proxy SNPs were used, selected based on high linkage disequilibrium (r² > 0.8) with the original SNPs to preserve the instrument's validity. We acknowledge that the impact of using proxy SNPs on the results can be significant and will include a discussion on the potential biases and limitations associated with this approach in the revised manuscript.

2. Results Interpretation:

o Association vs. Causation: The paper concludes a protective effect of amlodipine on GERD risk. However, the distinction between association and causation needs stronger emphasis, especially given the nature of MR studies.

o Response: We appreciate the reviewer's feedback on this critical point. While our Mendelian Randomization study suggests a potential causal relationship between amlodipine use and reduced GERD risk, we recognize that MR studies have inherent limitations and cannot establish causation definitively. We will revise the discussion section to emphasize the distinction between association and causation more clearly, acknowledging the assumptions and limitations of the MR approach, such as potential pleiotropy and unmeasured confounding.

3. Functional Analysis and Biological Pathways:

o Enrichment Analysis: The connection between identified pathways and GERD is inferred but not deeply explored. How calcium ion transport and related pathways specifically influence GERD pathophysiology should be more explicitly discussed so that readers may better understand it.

o Response: Thank you for pointing out the need for a deeper exploration of the identified pathways. Our enrichment analysis highlighted calcium ion transport and signalling pathways as key biological processes involved. These pathways are crucial in regulating smooth muscle tone, including the lower oesophageal sphincter (LES), which plays a significant role in GERD pathophysiology. We will expand the discussion to provide a more detailed explanation of how these pathways influence GERD development and the potential mechanisms through which amlodipine exerts its protective effects.

4. General

o Comparison with Other Medications: While amlodipine is highlighted, comparisons with other calcium channel blockers or antihypertensive medications are not well developed. This comparison could strengthen the argument for amlodipine's unique role.

o Response: We appreciate the reviewer’s suggestion regarding the comparison with other calcium channel blockers or antihypertensive medications. However, the scope of our study was specifically designed to focus on the potential protective effect of amlodipine against GERD. Due to the study’s design and the available data, we were not equipped to conduct a comprehensive comparison with other medications within this framework. Our study aimed to elucidate the genetic relationship between amlodipine use and GERD risk, utilizing a Mendelian Randomization approach. Expanding the analysis to include other medications would require additional data sets and potentially different methodological approaches, which were beyond the scope of the current study. We acknowledge that such comparisons could provide valuable insights and strengthen the argument for amlodipine's unique role. Therefore, we suggest that this could be an important direction for future research. Future studies that include direct comparisons between different classes of calcium channel blockers or antihypertensive drugs may further clarify the unique effects of amlodipine on GERD risk. Such research could help identify specific pharmacological properties of amlodipine that contribute to its protective effects, providing a more comprehensive understanding of its role in GERD management.

Minor Concerns

1. Writing and Presentation:

o Abstract and Introduction: The abstract is comprehensive, but the introduction could benefit from a clearer statement of the study's novelty and specific hypotheses.

o Response: Thank you for your constructive feedback. We agree that the introduction could better highlight the study's novelty and hypotheses. We will revise the introduction to explicitly state our research question and the novel aspects of our study, such as the application of a bidirectional MR approach and the construction of a ceRNA network to explore the genetic relationship between amlodipine use and GERD.

o Figures and Tables: The figures (e.g., funnel plots, enrichment analyses) are informative, but the legends should provide more contexts to aid interpretation. 

o Response: We appreciate the feedback on the figures and tables. We will revise the figure legends to include more detailed explanations, ensuring that they provide sufficient context for readers to interpret the data accurately. This will include descriptions of the methods used, the significance of the results, and any relevant biological implications.

2. Statistical Methods:

o Software and Packages: While the use of R and specific MR packages is mentioned, a more detailed description of the statistical methods and assumptions would help readers replicate the study.

o Response: Thank you for highlighting the need for a more detailed description of our statistical methods. We will expand the methods section to include a comprehensive description of the statistical software, and packages used, as well as the assumptions underlying our MR analyses. This will help ensure that the study can be replicated by other researchers.

o Sensitivity Analyses: The results of sensitivity analyses are presented, but the implications of findings like the MR-Egger intercept and Cochran’s Q-test are not explained.

o Response: We acknowledge that the implications of our sensitivity analyses require further elaboration. The MR-Egger intercept and Cochran’s Q-test are crucial in assessing pleiotropy and heterogeneity, respectively. We will revise the manuscript to explain the significance of these tests and how they support the robustness and validity of our findings.

3. Technical Accuracy:

o Genetic Terms and Definitions: Some genetic terms (e.g., linkage disequilibrium, pleiotropy) are used without definitions. Including brief explanations would benefit readers who are not geneticists.

o Response: We appreciate the suggestion to provide definitions for genetic terms. We will include brief explanations of key genetic terms, such as linkage disequilibrium and pleiotropy, to enhance the manuscript's accessibility for readers from diverse backgrounds.

o Gene Lists and Databases: The sources of gene lists and databases used for functional analysis are cited, but the process of selecting and validating these resources should be clearer.

o Response: Thank you for pointing out the need for clarity in our selection and validation process for gene lists and databases. We will revise the manuscript to provide a detailed description of the criteria and methods used to select and validate these resources, ensuring transparency and reproducibility of our analysis.

---

## [Editor Report · Decision Letter 1]

20 Aug 2024

Genetic Evidence for Amlodipine's Protective Role in Gastroesophageal Reflux Disease: A Focus on CACNB2

PONE-D-24-22095R1

Dear Dr. Shao

We’re pleased to inform you that your manuscript has been judged scientifically suitable for publication and will be formally accepted for publication once it meets all outstanding technical requirements.

Kind regards,

Asif Jan, Ph.D

Academic Editor

PLOS ONE
---

## [Editor Report · Acceptance letter]

18 Sep 2024

PONE-D-24-22095R1 

PLOS ONE

Dear Dr. Shao, 

I'm pleased to inform you that your manuscript has been deemed suitable for publication in PLOS ONE. Congratulations! Your manuscript is now being handed over to our production team.

Kind regards, 

on behalf of

Dr. Asif Jan 

Academic Editor

PLOS ONE